# The Impact of Non-Andrological Medications on Semen Characteristics, Oxidative Stress and Inflammatory Parameters

**DOI:** 10.3390/medicina59050903

**Published:** 2023-05-08

**Authors:** Gerardo Salerno, Marina Borro, Vincenzo Visco, Soraya Olana, Francesca Gargano, Salvatore Raffa, Virginia Zamponi, Camilla Mancini, Antongiulio Faggiano, Maurizio Simmaco, Rossella Mazzilli

**Affiliations:** 1Laboratory of Clinical Biochemistry, Sant’Andrea University Hospital, 00189 Rome, Italy; marina.borro@uniroma1.it (M.B.); francesca-garg@hotmail.it (F.G.); maurizio.simmaco@uniroma1.it (M.S.); 2Department of Neurosciences, Mental Health and Sensory Organs (NESMOS), Sapienza University of Rome, 00189 Rome, Italy; 3Medical Genetics and Advanced Cellular Diagnostics Unit, Sant’Andrea University Hospital & Department of Clinical and Molecular Medicine, Sapienza University of Rome, 00189 Rome, Italy; vincenzo.visco1@uniroma1.it (V.V.); salvatore.raffa@uniroma1.it (S.R.); 4Unit of Andrology and Endocrinology, Department of Clinical and Molecular Medicine, Sapienza University of Rome, 00189 Rome, Italy; soraya.olana@gmail.com (S.O.); virginia.zamponi@uniroma1.it (V.Z.); endocrinologiastudycoordinator@gmail.com (C.M.); antongiulio.faggiano@uniroma1.it (A.F.); rossella.mazzilli@uniroma1.it (R.M.)

**Keywords:** male infertility, medications, inflammation, cytokine, interleukin

## Abstract

*Background and Objectives*: The aim of this study was to evaluate the impact of medications on oxidative stress, inflammatory biomarkers and semen characteristics in males with idiopathic infertility. *Materials and Methods*: In this observational case-control clinical study, 50 men with idiopathic infertility were enrolled, of whom 38 (the study group) were on pharmacological treatment and 12 made up the control group. The study group was clustered according to the medications (Group A: anti-hypertensive, *n* = 10; Group B: thyroxine, *n* = 6; Group C: non-steroidal anti-inflammatory drugs, *n* = 13; Group D: miscellaneous, *n* = 6; Group E: lipid-lowering drugs, *n* = 4). Semen analyses were performed according to WHO 2010 guidelines. Interleukins (IL)-10, IL-1 beta, IL-4, IL-6, Tumor Necrosis Factor- alpha (TNF-alpha) and IL-1 alpha were determined using a solid-phase sandwich immunoassay. The diacron reactive oxygen metabolites, d-ROMs test, was performed by means of a colorimetric determination of reactive oxygen metabolites and measured with a spectrophotometer. Beta-2-microglobulin and cystatin-C were measured with an immunoturbidimetric analyzer. *Results*: No differences between the study and control groups for age and macroscopic and microscopic semen characteristics were found, nor were any differences found after clustering according to the drug categories. IL-1 alpha and IL-10 were significantly lower in the study group compared with the control group; IL-10 was significantly lower in groups A, B, C and D compared with the control group. Furthermore, a direct correlation between IL-1 alpha, IL-10 and TNF-alpha and leukocytes was found. *Conclusions*: Despite the sample size limitations, the data suggest a correlation between drug use and activation of the inflammatory response. This could clarify the pathogenic mechanism of action for several pharmacological classes on male infertility.

## 1. Introduction

Pharmacological treatments used by male partners of infertile couples could negatively influence seminal parameters, affecting the sperm number as well as the sperm motility [1]. Medications can affect the male reproductive system through central hormonal effects, sexual function, effects on sperm function or direct gonadotoxic effects [2,3]. It is well known that chemotherapeutic agents could have a direct gonadotoxic effect. Moreover, we recently found sperm progressive motility to be significantly reduced in subjects treated with psychotropic drugs [3]. Other medications have been less investigated. For instance, the use of sulfasalazine appears to be the cause of oligo-astheno-teratospermia and infertility in many treated men [4]. Similarly, acetylsalicylic acid and non-steroidal anti-inflammatories administered chronically can affect sperm quality by decreasing sperm count, motility, vitality and morphology [5]. In addition, statins can affect the sperm parameters and the seminal fluid composition of healthy men [6].

The analysis of inflammation biomarkers (oxidative stress and cytokines) represents one of the new tools to investigate the causes of male infertility [7].

Interleukin 1 alpha (IL-1a), Interleukin 1 beta (IL-1b), Interleukin 4 (IL-4), Interleukin 6 (IL-6), Interleukin 10 (IL-10) and tumor necrosis factor-alpha (TNF-alpha) are produced physiologically in the male gonads and appear as constituents of seminal plasma [8]. The precise role and regulation of cytokines in the genital tract is still under investigation. The greatest number of cytokines available in the male genital tract is produced by the testes.

More specifically, IL-1 and IL-6 are produced by testicular macrophages, by Leydig and Sertoli cells [9]. Furthermore, IL-4 and IL-10 are synthesized by T-helper 2 cells [10] and are involved in inflammatory and humoral immunological activity [11]. The concentrations of these cytokines were found to be significantly lower in the seminal plasma of the infertility subjects compared with the control group [12]. TNF-α is present in semen and also secreted by immune cells, mesenchymal cells, Sertoli cells and spermatogonia [13]. Together with IL-6 and IL-10, TNF-α represents a pro-inflammatory cytokine.

IL-1a and IL-1b are both produced as a 31-kd precursor and secreted as a 17-kd peptide [14]. IL-1α, mainly produced by the Sertoli or spermatogenic cells during the maturation cycles of the seminiferous epithelium, performs a crucial role in the control of testicular functions [7,15]; it appears to be a marker of testicular function. IL-1 beta induces an increase in reactive oxygen species (ROS) production, causing a reduction in sperm motility and an increase in apoptosis [16]. In addition, beta-2-microglobulin has been detected in high levels in seminal fluid, and the concentration is related to the sperm count [17,18]. Furthermore, cystatin C (Cys C) showed a significant increase in infertile men whose fertility was affected by varicocele [19].

Finally, recent evidence has highlighted that the d-ROMs test could be considered a useful tool for evaluating the expression of oxidative stress and free-radical-derived compounds in semen [20].

However, to our knowledge, no studies are currently available evaluating the implication of pharmacological treatments on the inflammatory parameters of seminal fluid. On the other hand, some research suggests the implication of these parameters in the onset of chronic inflammatory conditions related to pathological conditions (i.e., varicocele) [21].

The aim of this study was to evaluate the impact of pharmacological treatment on semen characteristics, oxidative stress, and inflammatory parameters in the male partners of infertile couples with idiopathic infertility.

## 2. Materials and Methods

### 2.1. Patients

This observational case-control clinical study included patients referred to the Endocrinology–Andrology Unit of Sant’Andrea Hospital, Sapienza University of Rome, from October 2017 to June 2018. Eligible patients were invited to participate in the study and, in the case of acceptance, were informed about the purposes of the study and asked to sign a written consent.

Inclusion criteria were as follows: (a) male partner of infertile couple with idiopathic infertility, in absence of known female factor; (b) aged between 18 and 45 years; (c) on pharmacological treatment for at least three months before the enrollment; (d) having a hormonal profile within the reference ranges. Exclusion criteria were the following: (a) use of psychotropic drugs (due to a previous study of ours on this topic [3]) and (b) any known cause of infertility, such as previous cycles of chemotherapy or radiotherapy, traumas, orchitis, funicular torsions and cryptorchidism, genetic alterations (i.e., karyotype alteration, Y chromosome microdeletion, cystic fibrosis mutation), hypergonadotropic or hypogonadotropic hypogonadism, non-compensated hypothyroidism/hyperthyroidism and hyperprolactinemia, obstructive azoospermia, severe varicocele (i.e., grade III varicocele), and other andrological diseases with a negative impact on spermatogenesis.

A total of 12 healthy age-matched men, who had conceived in the previous 6 months, were included in the control group. None of them used drugs.

All patients included in the study were evaluated as follows: (i) andrological clinical and physical examination; (ii) recording of demographic characteristics and use of any kind of medication; (iii) semen sample collection.

Patients were clustered according to the pharmacological treatment used: Group A: anti-hypertensive (amlodipine, angiotensin-II-receptor antagonists-sartans); Group B: levothyroxine; Group C: non-steroidal anti-inflammatory drugs (mesalazine, acetylsalicylic acid); Group D: miscellaneous (dimethyl fumarate, acenocoumarol, ursodeoxycholic acid); Group E: lipid-lowering agents (statins).

### 2.2. Semen Analysis

Each participant provided a semen sample which was collected by masturbation after sexual abstinence of between 2 and 7 days. Semen analysis was carried out according to 2010 WHO guidelines.

The sample container was placed in an incubator (37 °C) for 30–60 min. The physical and chemical characteristics of the seminal liquid were then evaluated (appearance, pH, liquefaction and viscosity—namely macroscopic characteristics). Furthermore, sperm concentration (10^6^/mL), total spermatozoa number (*n* × 10^6^/ejaculate), total motility and progressive motility (%), morphology (% abnormal forms) and round cells (*n* × 10^6^/mL) were evaluated under the optical microscope—namely microscopic characteristics (WHO 2010).

### 2.3. Cytokine Analysis in Seminal Plasma

Semen samples were centrifuged at 1500× *g* (NEYA 10R centrifuge, Remi Elektrotechnir Ltd., Vasai 401-208, India) for 10 min, and an aliquot of 350 µL of seminal plasma was stored at −80 °C until processing. Interleukin IL-4, IL-6, IL-10, TNF-α, IL-1α and IL-1β were determined using a human cytokine/chemokine magnetic bead panel A (Milliplex MAP kit, Millipore Corp., Billerica, MA, USA) and measured using a Magpix Luminex instrument and Xponent software (version 4.2, Luminex Corp, Austin, TX, USA) according to the manufacturer’s instructions. Samples of seminal plasma were analyzed undiluted. Luminex multiplex bead immunoassays are solid-phase sandwich immunoassays. Beads of defined spectral properties conjugated to analyte-specific capture antibodies and samples (including standards of known analyte concentration, control specimens and unknowns) are pipetted into the wells of a filter-bottom microplate and incubated. During this first incubation, analytes bind to the capture antibodies on the beads. After washing the beads, analyte-specific biotinylated detector antibodies are added and incubated with the beads. During this second incubation, the analyte-specific biotinylated detector antibodies recognize their epitopes and bind to the appropriate immobilized analytes. After removal of excess biotinylated detector antibodies, streptavidin conjugated to the fluorescent protein, R-phycoerythrin (Streptavidin-RPE), is added and incubated. During this final incubation, the Streptavidin-RPE binds to the biotinylated detector antibodies associated with the immune complexes on the beads, forming a four-member solid-phase sandwich. After washing to remove unbound Streptavidin-RPE, the beads are analyzed with the Luminex instrument. By monitoring the spectral properties of the beads and the amount of associated R-phycoerythrin (RPE) fluorescence, the concentration of one or more analytes can be determined.

### 2.4. Oxidative Stress Levels and Biomarkers Analysis in Seminal Plasma

The d-ROMs values were obtained using a colorimetric determination of reactive oxygen metabolites (d-ROMs test, Diacron, Italy COD. MC 002) and measured using a Biochrom Libra S12 UV/Vis spectrophotometer (Biochrom Ltd., Unit 7, Enterprise Zone 3970 Cambridge Research Park, Beach Drive, Waterbeach, Cambridge, UK, CB25 9PE) according to the manufacturer’s instructions. The d-ROMs test is based on Fenton’s reaction and consists of two steps. In the first step, hydroperoxides (ROOH) from a biological sample react with iron (released from plasma proteins by an acid buffer—the R2 reagent) and generate alkyl(R-O*) and peroxyl(R-OO*) radicals. In the second step, the alkyl and peroxyl radicals oxidize an alkyl-substituted aromatic amine A-NH2 (solubilized in a chromogenic mixture—the R1 reagent) and generate its oxidized form ([A-NH2*])+, a pink colored compound. The color change that occurs as a result of oxidation of the aromatic amine A-NH2 is photometrically quantifiable by means of a spectrophotometer.

Beta-2-microglobulin and cystatin-C were measured with an Optilite beta-2-microglobulin kit (LK043.OPT (The Binding Site Group Ltd., Birmingham, UK)) and an Optilite cystatin-C kit (LK048.OPT (The Binding Site Group Ltd., Birmingham, UK)) using an Optilite immunoturbidimetric analyzer (The Binding Site Group Ltd., Birmingham, UK). Samples were diluted 1:5 with Optilite Diluent 1 (IK709 (The Binding Site Group Ltd., Birmingham, UK)) and analyzed. The determination of the soluble antigen concentration using turbidimetric methods involves a reaction with specific antiserum to form insoluble complexes. When light is passed through the suspension formed, a portion of the light is transmitted and focused onto a photodiode by an optical lens system. The amount of transmitted light is indirectly proportional to the specific protein concentration in the test sample. The concentration is automatically calculated by reference to a calibration curve.

### 2.5. Statistical Analysis

All normally distributed variables were represented as mean and standard deviation (mean ± SD); variables that did not follow a normal distribution were represented as median with Q1 and Q3 quartiles or as number of cases and percentages (*n*, %) if categorical variables.

Subjects were stratified into two groups (study group and control group) and the variables were compared using the following statistical tests: the unpaired t-test for continuous variables if data were normally distributed or the corresponding Mann–Whitney U test for continuous variables in the case of non-normality and the χ^2^ test for categorical variables.

Differences between variables in the study subgroups (Group A, Group B, Group C, Group D, Group E) and the control group were determined using the Kruskal–Wallis nonparametric test with Dunn’s post hoc test.

All semen parameters were tested as dependent variables for multiple regression analysis. Only significant models were considered. Multiple regression analyses were performed using the white blood cells count (WBC count) as a dependent variable, and covariates were IL-10, IL-1 alpha, IL-1 beta, IL-4, IL-6, TNF-alpha, cystatin-C, beta-2-microglobulin and d-ROMs. The accuracy of the IL-10, IL-1 alpha, IL-1 beta, IL-4, IL-6, TNF-alpha, cystatin-C, beta-2-microglobulin and d-ROMs for distinguishing the presence of corpuscles and zones was assessed from the receiver operating characteristic (ROC). The results were expressed as area under the curve (AUC). Correlation analyses were performed using Spearman’s rank correlation.

Multiple regression, Spearman’s rank correlation and ROC curve were performed in the study group, and *p* value < 0.05 was considered statistically significant.

All statistical analyses were performed using R program version 4.0.3 (2020-10-10)—“Bunny-Wunnies Freak Out” (Copyright 2020, R Foundation for Statistical Computing, Vienna, Austria. URL https://www.R-project.org/).

We calculated the sample size using the variable IL-6 evaluated in the seminal fluid of the infertile group (22). Assuming a power(1-ß) of 95% with alpha equal to 5%, the sample size was 27 subjects.

## 3. Results

A total of 110 male partners of infertile couple were evaluated. Of these, 72 subjects were excluded because they did not meet the inclusion criteria (8 patients used psychotropic drugs, 10 patients did not used any type of medications and 54 patients were affected by an already-known cause of infertility). This left a final sample of 38 subjects (mean ± SD age: 38.0 ± 8.22 years), divided as follows: Group A: anti-hypertensive, *n* = 10; Group B: levothyroxine, *n* = 6; Group C: non-steroidal anti-inflammatory drugs, *n* = 13; Group D: miscellaneous, *n* = 6; Group E: lipid-lowering agents, *n* = 4. The population was compared with a control group (*n* = 12 patients, mean ± SD age: 39.6 ± 6.9 years).

The two groups (total drugs group and control group) did not differ in respect of age and macroscopic and microscopic semen characteristics (Table 1). Moreover, no differences were observed after clustering in subgroups according to type of drugs, considering both macroscopic and microscopic semen characteristics (Table 2).

Considering seminal plasma biomarkers of inflammation, IL-1 alpha and IL-10 were significantly lower in the total drugs group compared to the control group (median 20.2 vs. 42.5 ng/mL; *p* = 0.04, and median 22.1 vs. 43.0 ng/mL; *p* < 0.001, respectively) (Figure 1b). No significant differences were observed between the two groups for IL-1 beta, IL-4, IL-6 and TNF-alpha, as well as for d-ROMs, beta-2-microglobulin and cystatin-C (*p* not significant (*p* = n.s.)) (Table 1).

After clustering of patients, IL-10 was significantly lower in groups A, B, C and D compared with the control group (median 22.9, 20.2, 21.3, and 22.7 vs. 43.0 ng/mL, respectively; *p* = 0.003) (Figure 1a).

No significant differences were observed among the subgroups for IL-1 alpha, IL-1 beta, IL-4, IL-6 and TNF-alpha, as well as for d-ROMs, beta-2-microglobulin and cystatin-C (*p* = n.s.) (Table 3).

A multiple regression was calculated to predict the WBC count. IL-10 (estimate = 0.014, *p*-value = 0.012) and IL 1- alpha (estimate = 0.003, *p*-value = 0.010) were positively associated with the WBC count, while d-ROMs (estimate = −0.0009, *p*-value = 0.041) was negatively associated (Table 4).

In the study group, the ROC curves, correlations and multiple regression were evaluated (Table 5).

The ROC curves to predict the event ‘’corpuscles’’ (Figure 1f) were significantly predictive for IL-4 (AUC = 64.78, CI 95% = 46.64–82.93) and d-ROMs (AUC = 62.61, CI 95% = 43.99–81.22) (Figure 1), while for the event ‘’zones’’ (Figure 1e) compared with all the biomarkers analyzed, the curves were significantly predictive for variables IL-1 alpha (AUC = 71.72, CI 95% = 54.68–88.75) and TNF-alpha (AUC = 71.89, CI 95% = 52.00–91.77).

Of interest among the correlations examined were the statistically significant direct relationships (*p*-value <0.05) between the following: WBC count and IL-1 alpha, IL-10 and TNF-alpha respectively; total motility and d-ROMs; progressive motility and d-ROMs; volume and d-ROMs; and age and d-ROMs. Statistically significant inverse relationships were found between sperm concentration and IL-4 and total sperm number and beta-2 (Figure 1d).

## 4. Discussion

Adverse effects of drugs on the male reproductive system are often inadequately investigated. Notwithstanding, several medications can affect male fertility through different mechanisms [2]. For instance, antihypertensive and psychotropic agents can affect sexual function and hormonal parameters, while chemotherapeutic agents could have a direct gonadotoxic effect. In any case, the pathogenic mechanism is not completely known for several pharmacological classes. The analysis of inflammation parameters (oxidative stress and cytokines) represents one of the new tools to investigate the causes of male infertility [7,21].

In this regard, Attia et al. assessed the potential role of proinflammatory cytokines (TNF-α and IL-6) in the seminal plasma of subfertile men [7]. Having failed to determine a diagnostic value in male infertility, they suggested, however, that the evaluation of pro-inflammatory cytokines in the semen may improve the diagnosis of male infertility [7]. However, to our knowledge, no studies are currently available evaluating the implication of medications on cytokines and other inflammatory parameters of the seminal fluid.

In the present study, we evaluated 50 subjects with idiopathic infertility, and we analyzed the effect of various medications on semen parameters and on IL-10, IL-1 alpha, IL-1 beta, IL-4, IL-6, TNF-alpha, cystatin-C, beta-2-microglobulin and d-ROMs, in order to assess the role of pharmacological treatments on inflammation parameters. We excluded patients with already-known causes of infertility, with the aim of minimizing interference related to specific pathologies and not to pharmacological treatments. In fact, in this regard, Paoli et al. observed that varicocele patients showed a significant reduction in IFN-γ, IL-6 and TNF-α and an increase in IL-10, suggesting the presence of a chronic inflammatory condition, which could be a cause of the semen alteration [21].

We found that IL-1 alpha and IL-10 were significantly lower in the total drugs group compared with the control group. The result was confirmed for IL-10 also, after clustering the total drugs group in subgroups.

On the other hand, no significant differences were observed between the two groups for IL-1 beta, IL-4, IL-6 and TNF-alpha, as well as for d-ROMs, beta-2-microglobulin and cystatin-C.

The activities and levels of IL-10 in semen have been analyzed by several authors. Zhang et al. [12] observed that Il-10 was significantly reduced in infertile patients compared with the control group; there was also a difference in levels of IL-4 and IL-1 beta between the WBC and the non-WBC semen groups. Huleihel et al. [14] observed that IL-10 levels were significantly lower in the seminal plasma of patients with genital infection and oligoterato-asthenoazoospermia compared with that of fertile donors. Conversely, Seshadri et al. [22] found an increase of IL-10 in the asthenospermic, oligoasthenospermic and azoospermic groups compared with the normospermic group, thus assuming that the testis is probably not the site of its production; however, the authors also found a correlation between IL-10 and other proinflammatory cytokines in the oligospermic, asthenospermic, oligoasthenospermic and azoospermic groups, with increases also in IL-6, TNF-alpha and IL-8.

The role played by IL-10 is crucial in limiting inflammatory effects and in maintaining homeostasis. The biological activity sustained by IL-10 is mediated by heterodimeric IL-10 receptor expressed at varying degrees in different cell types, but macrophages and monocytes represent the first target. If the IL-10 deficiency is a long-term one, it can be harmful as the excessive production of proinflammatory cytokines could cause septic shock during bacterial, viral and fungal infections [23].

In this study, the significantly reduced levels of IL-10 in the drugs group could be due to its consumption in order to maintain homeostasis and avoid an increase in oxidative stress and biochemical and functional alterations of the seminal fluid. This hypothesis would be reinforced by the simultaneous reduction in the drugs group of IL-1 alpha levels with relative blocking of inflammatory activity.

In support of this speculation, we have observed how the levels of IL-1 beta, IL-6, TNF-alpha, Beta-2-microglobulin, cystatin-C and d-ROMs were comparable between the two groups, showing that there were no inflammatory alterations or evident oxidative activities. The levels of IL-4 were comparable between the two groups, and this would argue in favor of the fact that the drugs used would not determine a condition of evident inflammation but only a form of preventive containment inflammation exercised by IL-10. Evidence that the observed drug administration did not generate latent infertility is derived from the observation that none of the parameters of the classical analysis of the seminal fluid showed statistically significant differences between the two groups, except in relation to the pH.

IL-1 alpha can have effects as a paracrine mediator interacting positively with hormones of the GH/IGF-I system adjusting testicular cell functions [24]. The direct correlation between IL-1 alpha and IL-10 could be due to the testicular impairment related to inflammation.

Furthermore, among the correlations examined, we observed a direct relationship between IL-1 alpha and IL-10, between TNF-alpha and WBC and between d-ROMs and sperm motility, volume and age. Conversely, we found a statistically significant inverse relationship between IL-4 and sperm concentration and between beta-2 and total sperm number.

As regards the positive correlation between d-ROMs and sperm motility that we found, considering that the variations in the d-ROMs test values were not statistically significant in the two groups, and that it was not present in the controls, this could derive from the energetic metabolism produced by the process of mitochondrial phosphorylation and cytoplasmic glucose-6-phosphate dehydrogenase (G-6-PDH) contained in spermatozoa [25]. The inverse relationship between beta-2 and total sperm number was in accordance with previous studies. Ulèová-Gallová et al. [26] showed that the concentration of beta-2-microglobulin was significantly higher in 14% of infertile men analyzed in the study.

From the data from the multiple regression, with respect to the dependent variable WBC, we found Il-10, IL-1 alpha and d-ROMs test values to be significant independent factors. The model allowed us to show how both IL-10 and Il-1 alpha act in synergy and respond to the variation in WBC, while the d-ROMs, being negatively correlated, undergo the effects of IL-10. Lastly, the ROC curves performed with respect to zones showed IL-1 alpha and TNF-alpha to be significantly predictive factors, while with respect to corpuscles, IL-4 and d-ROMs as the dependent variables proved to be predictors.

The ROC analysis links the predictivity of three inflammatory factors (IL-1 alpha, TNF-alpha and d-ROMs) with cellular elements that are present in reduced numbers in the seminal fluid and that increase in the case of infections [27,28].

Finally, Candenas et al. evaluated the composition and proteome of seminal plasma as a source of biomarkers of male infertility. Interestingly, the authors carried out an examination of the biomarkers of male infertility described in the literature and stratified them by pathology and functional changes, namely azoospermia, primary and secondary infertility, asthenozoospermia, oligozoospermia, teratozoospermia and varicocele [29]. Our choice of criteria for the analysis of the proposed biomarkers was based on the particular characteristic of the study population. We evaluated the basic oxidative and inflammatory balance (through d-ROMs and cytokines), and we indicated the biomarkers used to characterize and support any alterations of the oxidative-inflammatory panel, which could easily be performed by the laboratory. In this regard, we aimed to understand a possible alteration induced by the therapy, common to all of the classes of drugs analyzed. Once the behavior of IL-10 and IL-1 alpha has been ascertained, the next goal will be to use more specific biomarkers to characterize these aspects.

The main limitations of this study are the small number of patients for each category, the heterogeneity of the treatment in the subgroups and the characteristics of the control group. We recruited healthy age-matched men, who had conceived in the previous 6 months and who were not taking any medications. However, to evaluate in more detail the effect exerted by the various classes of drugs, it could be useful in further studies to compare the population with subjects affected by a primary disorder, but not taking medications (e.g., patients with hypertension, but not taking anti-hypertensive drugs). On the other hand, the enrollment of subjects with these characteristics could be very difficult, considering that pharmacological therapies are undertaken rapidly to improve the perceived quality of life.

## 5. Conclusions

Our work represents a first attempt to understand how drugs belonging to different categories and used by patients can affect the properties of seminal fluid. Despite the sample size limitations, the data showed us how the various pharmacological therapies did not impact fertility, as the parameters of the seminal fluid were comparable with the controls.

We also found a decrease in IL-10 and IL-1 alpha in all subjects undergoing therapy. This behavior, which follows the rest of the proposed results, was attributed to a maintenance of homeostasis exerted by the cellular and humoral components present in the male reproductive system.

An analysis carried out on a greater number of patients will clarify in more detail the mechanisms underlying the results we have obtained.

## 6. Patients

This study involving human participants was reviewed and approved by Sapienza University of Rome, Study protocol: RM11916B8900AF26, Approval date: 26 June 2019. The patients/participants provided their written informed consent to participate in this study.

## Figures and Tables

**Figure 1 medicina-59-00903-f001:**
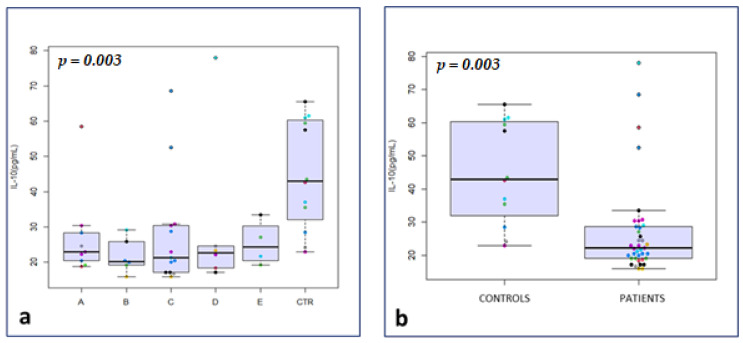
IL-10 in drug subgroups vs. The symbols in different colors showed individual observations. CTR (**a**); Il-10 in total drugs group vs. CTR (**b**); IL-1 alpha in total drugs group vs. CTR (**c**); matrix of Spearman’s correlation in the study group (**d**); ROC curves to predict the event “zones” in the study group (**e**); ROC curves to predict the event “corpuscles” in the study group (**f**).

**Table 1 medicina-59-00903-t001:** Baseline clinical and demographic characteristics and seminal plasma biomarkers in patients from total drugs group and control group (CTR group).

Baseline Characteristics				
		Total Drugs *n* = 38	CTR Group *n* = 12	*p*-Value
Age (years)		38.0 (8.2)	39.6 (6.9)	0.5860
pH		7.1 (0.1)	7.5 (0.1)	**0.0178**
Volume (mL)	mean (SD)	3.2 (1.9)	3.8 (1.9)	0.3728
Sperm concentration	mean (SD)	61.7 (36.5)	68.4 (35.8)	0.5988
Total sperm number	mean (SD)	234.0 (211.5)	224.3 (182.0)	0.8825
Total motility	mean (SD)	37.5 (16.2)	40.8 (10.3)	0.4316
Progressive motility	mean (SD)	35.7 (17.5)	39.4 (11.7)	0.433
Abnormal morphology	mean (SD)	78.1 (7.2)	75.9 (3.9)	0.2056
WBC count	mean (SD)	0.7 (0.4)	0.6 (0.3)	0.7551
Viscosity	*n* (%)	3 (7.8)	0 (0.0)	0.99
Aspect	*n* (%)	3 (7.9)	3 (25.0)	0.141
Corpuscles	*n* (%)	15 (39.5)	5 (41.7)	0.99
Zones	*n* (%)	11 (28.9)	6 (50.0)	0.294
**Seminal Biomarkers**				
IL-1 alpha, pg/mL	Median	20.2	42.5	**0.043 ***
	Q1, Q3	10.6, 31.9	22.1, 78.4	
IL-1 beta, pg/mL	Median	2.7	1.8	0.206 *
	Q1, Q3	1.9, 3.4	1.6, 4.0	
IL-4, pg/mL	Median	16.7	15.5	0.064 *
	Q1, Q3	15.8, 19.5	14.3, 17.5	
IL-6, pg/mL	Median	11.5	11.5	0.973 *
	Q1, Q3	7.4, 17.7	8.3, 15.1	
IL-10, pg/mL	Median	22.3	43.000	**0.0003 ***
	Q1, Q3	33.8, 22.2	33.8, 59.9	
TNF-alpha, pg/mL	Median	11.5	9.6	0.2170 *
	Q1, Q3	8.9, 14.2	8.5, 11.7	
d-ROMs, U.CARR	Median	363.2	405.1	0.3940 *
	Q1, Q3	322.8, 427.6	347.8, 456.6	
Beta-2-microglobulin mg/L	Median	40. 5	38.3	0.9910 *
	Q1, Q3	29.5, 51.9	31.4, 47.9	
Cystatin-C, mg/L	Median	37.3	46.3	0.2608 *
	Q1, Q3	25.7, 45.9	31.9, 53.5	

* Mann–Whitney test. Significant *p*-values were shown in bold.

**Table 2 medicina-59-00903-t002:** Baseline clinical and demographic characteristics of patients from different subgroups.

Baseline Characteristics for Subgroups							
	Group A *n* = 9	Group B *n* = 6	Group C *n* = 13	Group D *n* = 6	Group E *n* = 4	Group CTR *n* = 12	* *p*-Value
Age (years), mean (SD)	39.2 (4.5)	35.8 (3.5)	36.4 (9.0)	37.8 (13.2)	45.9 (5.0)	39.6 (6.9)	0.833
pH, mean (SD)	7.4 (0.1)	7.4 (0.1)	7.5 (0.1)	7.4 (0.1)	7.4 (0.1)	7.5 (0.1)	0.05
Volume (mL), mean (SD)	3.4 (1.5)	3.2 (1.0)	3.4 (1.6)	2.7 (1.2)	2.9 (1.9)	3.8 (1.9)	0.82
Sperm concentration, mean (SD)	62.4 (36.5)	73.8 (33.5)	63.5 (40.8)	46.2 (39.4)	62.8 (30.5)	68.4 (35.8)	0.864
Total sperm number, mean (SD)	317.6 (360.8)	231.8 (129.2)	221.8 (161.9)	144.9 (143.8)	243.2 (145.3)	224.3 (182.0)	0.783
Total motility, mean (SD)	34.6 (15.1)	43.6 (9.5)	39.6 (16.6)	33.3 (23.3)	35.3 (16.2)	40.8 (10.3)	0.818
Progressive motility, mean (SD)	31.6 (17.6)	41.6 (11.3)	38.2 (24.1)	32.8 (24.1)	32.8 (15.9)	39.4 (11.7)	0.837
Abnormal morphology, mean (SD)	78.8 (6.8)	75.4 (3.1)	78.2 (7.4)	81.3 (11.5)	75.3 (2.9)	75.9 (4.0)	0.573
WBC count, mean (SD)	0.8 (0.5)	04.5 (0.3)	0.6 (0.4)	0.7 (0.4)	0.6 (0.4)	0.6 (0.2)	0.880
Viscosity, *n* (%)	1 (11.0)	0 (0.0)	1 (7.7)	1 (16.7)	0 (0.0)	0 (0.0)	0.741
Aspect, *n* (%)	0 (0.0)	0 (0.0)	1 (7.7)	1 (16.7)	1 (25.0)	3 (25.0)	0.401
Corpuscles, *n* (%)	5 (55.5)	2 (33.3)	4 (37.8)	2 (33.3)	2 (50)	5 (41.7)	0.986
Zones, *n* (%)	4 (44.4)	0 (0.0)	5 (38.5)	1 (16.7)	1 (25.0)	6 (50.0)	0.316

* Kruskal–Wallis test. Group A = anti-hypertensive (amlodipine, losartan), Group B = levothyroxine, Group C = mesalazine, acetylsalicylic acid, Group D = various (dimethyl fumarate, acenocoumarol, ursodeoxycholic acid), Group E = statins.

**Table 3 medicina-59-00903-t003:** Analysis of semen biomarkers for subgroups.

Seminal Biomarkers for Subgroups								
		Group A *n* = 9	Group B *n* = 6	Group C *n* = 13	Group D *n* = 6	Group E *n* = 4	Group CTR *n* = 12	* *p*-Value
IL-1 alpha, pg/mL	median	23.2	20.2	20.2	17.1	18.0	42.5	0.380
	Q1, Q3	20.2, 31.4	10.2, 25.5	14.2, 32.1	8.3, 33.5	14.5, 25.7	22.1, 78.4	
IL-1 beta, pg/mL	median	2.7	2.3	2.7	4.8 **	3.0	1.8	0.140
	Q1, Q3	2.7, 4.1	1.4, 2.7	2.0, 3.4	2.9, 7.2	2.7, 3.7	1.6, 9.0	
IL-4, pg/mL	median	16.7	15.5	16.7	17.5	17.2	15.5	0.287
	Q1, Q3	16.4, 20.0	14.5, 16.7	15.7, 19.6	16.9, 18.8	16.7, 18.1	14.3, 17.5	
IL-6, pg/mL	median	14.2	10.5	11.0	7.3	20.1	11.5	0.597
	Q1, Q3	7.1, 17.0	5.5, 16.6	7.9, 15.1	5.7, 28.4	14.6, 24.5	8.3, 15.1	
IL-10, pg/mL	median	22.9	20.2	21.3	22.7	24.3	43.0	**0.003**
	Q1, Q3	20.4, 28.3	19.4, 24.4	17.1, 30.3	19.3, 24.2	21.0, 28.6	33.8, 59.9	
TNF-alpha, pg/mL	median	13.3	10.085	10.558	11.473	13.723	9.630	0.329
	Q1, Q3	9.6, 14.2	8.7, 11.5	9.6, 11.5	9.2, 13.7	10.8, 17.7	8.5, 14.2	
d-ROMs, UCARR	median	389.7	569.9	383.8	347.6	386.8	405.1	0.542
	Q1, Q3	341.2, 448.5	326.1, 817.1	325.0, 426.5	257.7, 358.1	333.1, 448.2	314.7, 456.6	
Beta-2-microglobulin mg/L	median	36.2	31.2	46.9	44.6	38.0	38.3	0.652
	Q1, Q3	23.5, 72.9	25.9 43.6	39.3, 51.8	32.4, 65.9	26.6, 48.3	31.4, 47.9	
Cystatin-C, mg/L	median	27.1	34.2	45.7	38.2	31.1	46.3	0.481
	Q1, Q3	21.3, 58.7	30.5, 40.8	28.1, 53.1	26.7, 43.4	17.8, 41.9	32.0, 53.5	

* Kruskal–Wallis test; ** Dunn’s post hoc test, *p* < 0.05 vs Group CTR. Significant *p*-values were shown in bold.

**Table 4 medicina-59-00903-t004:** Multiple regression in the study group (dependent variable WBC count).

Multiple Regression (Drugs Patients *n* = 38)			
Dependent Variable = WBC Count F-Statistic: 4.553 on 9 and 26 DF, *p*-Value: 0.001133			
	Estimate	Std. Error	*p*-Value
Intercept	1.1196493	0.4453493	**0.0185**
IL-10	0.0141850	0.0052892	**0.0126**
IL-1 alpha	0.0031437	0.0011404	**0.0105**
Beta-2-microglobulin	0.0049970	0.0040362	0.2268
IL-1 beta	−0.0510248	0.0349584	0.1564
IL-4	−0.0275752	0.0184966	0.1480
IL-6	0.0012377	0.0043129	0.7764
TNF-alpha	−0.0138462	0.0182907	0.4559
Cystatin-C	−0.0006115	0.0036349	0.8677
d-ROMs	−0.0008977	0.0004170	**0.0408**

Significant *p*-values were shown in bold.

**Table 5 medicina-59-00903-t005:** Curve ROC analysis: dependent variables zones and corpuscles.

ROC CURVE			
Dependent Variable = Zones			
	AUC	CI 95%	*p*-Value
IL-1 alpha	71.7	54.7–88.8	**<0.05**
IL-1 beta	58.1	38.4–77.8	NS
IL-4	57.1	36.4–77.7	NS
IL-6	60.1	37.5–82.7	NS
IL-10	59.9	39.6–80.3	NS
TNF-alpha	71.9	52.0–91.8	**<0.05**
d-ROMs	56.6	34.0–79.2	NS
Beta-2-microglobulin	60.6	41.2–80.0	NS
Cystatin-C	55.7%	33.3–78.1	NS
Significant *p*-values were shown in bold			
**Dependent Variable = Corpuscles**			
	**AUC**	**CI 95%**	** *p* ** **-Value**
IL-1 alpha	52.9	33.8–72.1	NS
IL-1 beta	45.1	26.0–64.2	NS
IL-4	64.8	46.6–82.9	**<0.05**
IL-6	49.3	29.5–69.1	NS
IL-10	48.4	29.0–67.8	NS
TNF-alpha	54.1	34.6–73.5	NS
d-ROMs	62.6	44.0–81.2	**<0.05**
Beta-2-microglobulin	57.5	38.7–76.4	NS
Cystatin-C	59.6	40.8–78.3	NS

Significant *p*-values were shown in bold.

## Data Availability

Data are available after specific and appropriate request.

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
