# Peer review of "The Impact of Non-Andrological Medications on Semen Characteristics, Oxidative Stress and Inflammatory Parameters"

_medicina, 2023, doi:10.3390/medicina59050903_

Round 1

Reviewer 1 Report

The topic is interesting, and the design of the study is correct.

The sample size estimation was not mentioned in material and methods.

It is not clear which variable is considered for sample size estimation.

It is not clear which variable had normal distribution and not.

Author Response

Dear reviewer, thank you for your suggestions.

1) The sample size estimation was not mentioned in material and methods.

The sample size was preliminarily calculated using the work of  Qian L et al. 2011 (Qian L, Sun G, Zhou B, Wang G, Song J, He H. Study on the relationship between different cytokines in the semen of infertility patients. Am J Reprod Immunol. 2011 Aug;66(2):157-61. doi: 10.1111/j.1600-0897.2010.00980.x. Epub 2011 Jan 19. PMID: 21244563.).

2)It is not clear which variable is considered for sample size estimation.

The variable used to calculate the sample size was IL-6 evaluated in the seminal fluid of infertile Groups. Assuming a Power(1-ß) of 95% with alpha equal to 5%, the sample size was 27 subjects.

We added this information in the manuscript, as follows: “We calculate the sample size using the variable IL-6 evaluated in the seminal fluid of infertile Groups (22). Assuming a Power(1-ß) of 95% with alpha equal to 5%, the sample size was 27 subjects.”

3) It is not clear which variable had normal distribution and not.

In ‘’Statistical analysis’’ we have replaced ‘’All the variables analyzed were assessed by the Kolmogorov-Smirnov test. Data were summarized as mean and standard deviation (mean ±SD) if normally distributed, median with Q1 and Q3 quartile if not normally distributed or as number of cases and percentages (n, %) if categorical variables’’ with ‘’All normally distributed variables were represented as mean and standard deviation (mean ±SD); variables that did not follow a normal distribution were represented as median with Q1 and Q3 quartile or as number of cases and percentages (n, %) if categorical variables’’

Reviewer 2 Report

This manuscript presents an original and interesting approach to estimate possible impact of several medication on components of seminal plasma related to oxidative stress, inflammatory biomarkers and semen characteristics in patients diagnosed with idiopathc infertility. It presents clearly the methodology implemented and findings. The major default in this manuscript is  the limited number of patients included, in each subgroup prohibiting to estimate the real value of the observations. No calculations of power analysis are presented. I would suggest the authors to resubmit the paper after evaluating the real number of patients needed in each group and the control, that are needed to find differences with real statistical significance.

Author Response

Dear reviewer, thank you for your suggestions and comments.

1) This manuscript presents an original and interesting approach to estimate possible impact of several medication on components of seminal plasma related to oxidative stress, inflammatory biomarkers and semen characteristics in patients diagnosed with idiopathc infertility. It presents clearly the methodology implemented and findings. The major default in this manuscript is  the limited number of patients included, in each subgroup prohibiting to estimate the real value of the observations. No calculations of power analysis are presented. I would suggest the authors to resubmit the paper after evaluating the real number of patients needed in each group and the control, that are needed to find differences with real statistical significance.

The sample size was preliminarily calculated using the work of  Qian L et al. 2011 (Qian L, Sun G, Zhou B, Wang G, Song J, He H. Study on the relationship between different cytokines in the semen of infertility patients. Am J Reprod Immunol. 2011 Aug;66(2):157-61. doi: 10.1111/j.1600-0897.2010.00980.x. Epub 2011 Jan 19. PMID: 21244563.).

The variable used to calculate the simple size was IL-6 evaluated in the seminal fluid. Assuming a Power(1-ß) of 95% with alpha equal to 5%, the sample size was 27 subjects.

We added this information in the manuscript, as follows: “We calculate the sample size using the variable IL-6 evaluated in the seminal fluid of infertile Groups (22). Assuming a Power(1-ß) of 95% with alpha equal to 5%, the sample size was 27 subjects.”

However, we defined the sample size as the limit of the study since there are conflicting data in the literature regarding the variation of some of the parameters analyzed in infertile patients.

Considering the novelty introduced in this work, which concerns the observation of the alteration of numerous biomarkers in patients using drugs, we think it is right to extend the conclusions we have drawn to a larger group of patients. We started from the enrollment of 100 subjects, which, as described, were reduced for various reasons.

The statistical differences that we found in the analysis are very significant, and considering the novelty of the phenomenon described, we think that with all the limitations mentioned, the results could be useful for the scientific community to characterize the phenomenon in more detail.

Surely, our purpose for the future is to increase the number of cases with a multi-center study, which we hope to be able to submit again to your journal

Round 2

Reviewer 2 Report

The concept of this observational, case-control, clinical study is interesting and valuable.

 The aim of this study was to evaluate the impact of various medications on oxidative stress, inflammatory biomarkers and semen characteristics in male patients with idiopathic infertility. The paper is clearly written, and the data are well presented and analyzed. As the study group included only patients with unexplained infertility, the various parameters of their semen analysis were comparable to the control group. Examining inflammation biomarkers (oxidative stress and cytokines) differences in some parameters in their seminal plasma related to inflammatory response, for example a decrease in IL-10 and IL-1alpha in all subjects undergoing therapy , was observed. However, these results should be interpreted with caution due to the small sample size of the study groups and the possible confounding factors that may have influenced the results, such as patients' weight, smoking and general health status, including the variable illnesses requiring the administration of the variable medications. I understand the complexity of recruiting patients with similar health problems, without medication that in theory should have served another group for comparison enabling to distinguish the role of the medications used from the background health problem. In the discussion these limitations should be better explained. Also, as there is a wide array of possible biomarkers that may be evaluated, as reviewed by Candenas and Chianese and the discussion may be enriched by explaining the specific choses made by the investigators, examining the variables included in their research.

Still the idea to evaluate the relationship between various and specific medication used and semen/ seminal fluid characteristics is a novel aspect of evaluating male infertility, and this manuscript contributes to the current relevant knowledgebase. Therefore, I recommend accepting it for publication following minor corrections.

Candenas L.  and Chianese  R.  Exosome Composition and Seminal Plasma Proteome: A Promising Source of Biomarkers of Male Infertility Int. J. Mol. Sci. 2020, 21, 7022; doi:10.3390/ijms21197022

Author Response

Dear reviewer, thank you for your suggestions. Below, the changes we made based on your comments

1)I understand the complexity of recruiting patients with similar health problems, without medication that in theory should have served another group for comparison enabling to distinguish the role of the medications used from the background health problem. In the discussion these limitations should be better explained

1)R: Thank you for this observation. Following your suggestion, we modified the limitations as follows:

“ The main limitations of this study are represented by the small number of patients for each category, the heterogeneity of the treatment in subgroups and the characteristics of the control group. Indeed, we recruited healthy age-matched men, who have conceived in the previous 6 months and who were not taking any medications. However, to evaluate in more detail the effect exerted by the various classes of drugs, it could be useful in further studies to compare the population with subjects with subjects affected by the primary disorder, but not taking medications (i.e. patients with hypertension, not taking anti-hypertensive drugs). On the other hand, the enrollment of subjects with these characteristics could be very difficult, considering that pharmacological therapies are undertaken rapidly to improve the perceived quality of life.”

2) Also, as there is a wide array of possible biomarkers that may be evaluated, as reviewed by Candenas and Chianese and the discussion may be enriched by explaining the specific choses made by the investigators, examining the variables included in their research.

R: we added the following sentences to the discussion section: “Finally, Candenas et al evaluated the composition and seminal plasma proteome as  source of biomarkers of male infertility. Interestingly, the authors carried out an examination of the biomarkers of male infertility described in the literature and stratified by pathology and functional changes as, azoospermia, primary and secondary infertility, asthenozoospermia, oligozoospermia, teratozoospermia and varicocele. Our choice criterion for the analysis of the proposed biomarkers was based on the particular characteristic of the study population. We evaluated the basic oxidative and inflammatory balance (through dROMs and cytokines) and we indicated the biomakers used to characterize and support any alterations of the oxidative-inflammatory panel, which could be easily performed by the laboratory. In this regard, we aimed to understand a possible alteration induced by the therapy, common to all of the classes of drugs analyzed. Once the behavior of IL-10 and IL-1alpha has been ascertained, the next goal will be to use more specific biomarkers to characterize these aspects.”
